# Protein Quality Control Activation and Microtubule Remodeling in Hypertrophic Cardiomyopathy

**DOI:** 10.3390/cells8070741

**Published:** 2019-07-18

**Authors:** Larissa M. Dorsch, Maike Schuldt, Cristobal G. dos Remedios, Arend F. L. Schinkel, Peter L. de Jong, Michelle Michels, Diederik W. D. Kuster, Bianca J. J. M. Brundel, Jolanda van der Velden

**Affiliations:** 1Department of Physiology, Amsterdam UMC, Vrije Universiteit Amsterdam, Amsterdam Cardiovascular Sciences, 1081 HV Amsterdam, the Netherlands; 2Sydney Heart Bank, Discipline of Anatomy, Bosch Institute, University of Sydney, 2006 Sydney, Australia; 3Department of Cardiology, Thoraxcenter, Erasmus Medical Center, 3015 GD Rotterdam, the Netherlands; 4Department of Cardiothoracic Surgery, Thoraxcenter, Erasmus Medical Center, 3015 GD Rotterdam, the Netherlands; 5Netherlands Heart Institute, 3511 EP Utrecht, the Netherlands

**Keywords:** cardiomyopathy, hypertrophy, sarcomere mutation, haploinsufficiency, poison polypeptide, protein quality control, heat shock proteins, autophagy, microtubules

## Abstract

Hypertrophic cardiomyopathy (HCM) is the most common inherited cardiac disorder. It is mainly caused by mutations in genes encoding sarcomere proteins. Mutant forms of these highly abundant proteins likely stress the protein quality control (PQC) system of cardiomyocytes. The PQC system, together with a functional microtubule network, maintains proteostasis. We compared left ventricular (LV) tissue of nine donors (controls) with 38 sarcomere mutation-positive (HCM_SMP_) and 14 sarcomere mutation-negative (HCM_SMN_) patients to define HCM and mutation-specific changes in PQC. Mutations in HCM_SMP_ result in poison polypeptides or reduced protein levels (haploinsufficiency, HI). The main findings were (1) several key PQC players were more abundant in HCM compared to controls, (2) after correction for sex and age, stabilizing heat shock protein (HSP)B1, and refolding, HSPD1 and HSPA2 were increased in HCM_SMP_ compared to controls, (3) α-tubulin and acetylated α-tubulin levels were higher in HCM compared to controls, especially in HCM_HI_, (4) myosin-binding protein-C (cMyBP-C) levels were inversely correlated with α-tubulin, and (5) α-tubulin levels correlated with acetylated α-tubulin and HSPs. Overall, carrying a mutation affects PQC and α-tubulin acetylation. The haploinsufficiency of cMyBP-C may trigger HSPs and α-tubulin acetylation. Our study indicates that proliferation of the microtubular network may represent a novel pathomechanism in cMyBP-C haploinsufficiency-mediated HCM.

## 1. Introduction

Hypertrophic cardiomyopathy (HCM) is a common familial cardiac disease with an estimated prevalence of 1:200 [1]. It is characterized by an asymmetrically hypertrophied, non-dilated left ventricle and an impaired diastolic function [2]. More than 1500 pathogenic mutations that most frequently encode proteins in the sarcomere have been identified. Previous studies have shown that sarcomere mutations alter myofilament function (reviewed recently in [3]). However, while the same sarcomere mutation is present in members of one family, the age of onset and disease severity vary greatly between patients, spanning from asymptomatic mutation carriers to symptomatic patients with severe cardiac remodeling [4]. This large clinical heterogeneity implies that HCM pathophysiology is more complex than the functional defects triggered by the gene mutation. Furthermore, pathogenic mutations were only found in ~50% of all HCM patients [4]. Moreover, a recent large international clinical HCM study showed that patients with a mutation in a sarcomere gene had a 2-fold higher risk of adverse outcomes (e.g., heart failure and atrial fibrillation) compared to patients with no known mutations [5]. The exact mechanisms underlying the difference in disease severity between sarcomere mutation-positive (HCM_SMP_) and mutation-negative (HCM_SMN_) patients are currently unknown.

A mechanism implicated in HCM progression is perturbed protein quality control (PQC) [6,7]. Cardiomyocytes contain a multilayered PQC system for maintaining proper protein conformation and for reorganizing and removing misfolded or aggregated (mutant) proteins [8]. The PQC system comprises heat shock proteins (HSPs), the ubiquitin-proteasome system (UPS), and autophagy [9]. HSPs are involved at all stages of sustaining protein homeostasis: during protein biosynthesis and maturation, in chaperoning the folding, in protection from environmental stress, in rearrangements of cellular macromolecules during the functional cycles of assembly and disassembly, and finally, in targeting proteins for degradation [10]. Terminally misfolded and aggregation-prone proteins are cleared by the two degradation systems, UPS and autophagy [11]. Furthermore, pathways of PQC are strongly linked to cell architecture, such as the microtubules network [12]. As sarcomere proteins are the most abundant proteins in the heart, maintenance of sarcomere structure and functions depends on functioning PQC mechanisms. 

Perturbations in PQC mechanisms may be aggravated by the presence of sarcomere gene mutations that may explain the difference in disease severity between HCM_SMP_ and HCM_SMN_ patient groups. Moreover, the proteotoxic burden on PQC may depend on the type of mutation: in HCM, gene mutations either result in a ‘poison polypeptide’ (an expressed protein containing a mutated sequence) or haploinsufficiency, where up-regulation of the wild-type protein levels failed to compensate for reduced total protein [13,14]. 

Here, we identified and quantified key PQC players, such as HSPs and degradation markers (ubiquitinated proteins, p62, LC3BII) and microtubule network markers (for example acetylation of α-tubulin) in a large set of myectomy samples from HCM_SMP_ and HCM_SMN_ patients to define if the presence of a sarcomere mutation differentially affects PQC. In addition, mutation-specific PQC changes were assessed by comparing HCM with missense mutations and *MYBPC3* (cardiac myosin-binding protein-C) mutations, which cause haploinsufficiency.

## 2. Materials and Methods

### 2.1. Septal Myectomy 

Cardiac tissue from the interventricular septum was obtained during myectomy surgery to relieve left ventricular (LV) outflow tract obstruction. Patients were selected for surgery at our HCM center on the basis of the following indications: (1) peak left ventricular outflow tract (LVOT) gradient ≥ 50 mmHg at rest or on provocation and (2) presence of unacceptable symptoms, despite maximally tolerated medications consisting of β-blocking agents and/or calcium channel blockers. The decision to perform surgery was made after the consensus of a heart team consisting of a cardiothoracic surgeon, an interventional cardiologist, and a cardiologist specialized in HCM care [15]. Hypertrophic obstructive cardiomyopathy was evident from the increased septal thickness with a mean (± SD) of 21 (± 6) mm and high LV outflow tract pressure gradient with a mean (± SD) of 66 (± 37) mmHg. Next-generation-sequencing was used to check 48 different HCM-associated genes for variants [16].

This study included 52 septal myectomy patients, as summarized in Table 1. A total of 38 HCM patients were tested positive for carrying a gene variant in one of the genes encoding for the sarcomeric protein. The ages of all 38 HCM_SMP_ patients ranged between 15 and 72, with a mean (± SD) of 49 (± 16) years, and 68% of these were males. Depending on the reported pathomechanism, this group was subdivided into haploinsufficiency (HI) and poison polypeptide (PP) subgroups. 

Seven patients of the HCM_HI_ group were identified as carrying the Dutch founder mutation (c.2373dupG) in *MYBPC3*, the gene encoding the A band protein, cardiac myosin-binding protein-C (cMyBP-C) and twelve patients carried other mutations in *MYBPC3*, of which three were missense and the remainder were truncations. Patient ages ranged between 21 and 71 years, with a mean (± SD) of 43 (± 15) years, and 79% of these were males. We collected 19 myectomy samples from HCM_PP_ patients carrying a missense mutation. Nine patients presented with mutations in the gene for the cardiac myosin heavy chain (*MYH7*), the second most commonly mutated gene in HCM patients. Four patients carried mutations in the gene for troponin I (*TNNI3*), three patients carried troponin T (*TNNT2*) mutations and two patients carried myosin light chain 2 (*MYL2*) mutations. The HCM_PP_ patients were aged between 15 and 72 years, with a mean (± SD) of 55 (± 15) years, and 58% of these were males. 

Our study included 14 patients with no known sarcomere gene mutation. HCM_SMN_ patient ages ranged between 22 to 74, with a mean (± SD) of age 57 (± 14), and 64% of these were males. Not all the myectomy samples were sufficiently large to perform all the analyses.

Finally, the 52 HCM samples were compared with nine samples of healthy donor hearts that were not used for heart transplantation, mostly because of poor tissue matching against patients in the heart failure clinic of St Vincent’s Hospital. While these controls had no history of cardiac disease, their validity as “controls” was based on their extensive use as such in publications from a wide range of other laboratories. We therefore assumed that these donors are representative of the diverse human population. Consistent with the HCM patients, these donors were 67% male, with a mean age (± SD) of 43 (± 14) years, ranging from 19 to 65. All donor hearts were perfused with ice-cold cardioplegia and transported on ice to the Sydney Heart Bank where ~1 g samples were snap frozen and stored in liquid nitrogen. The average values of the control cardiac samples (*n* = 9) are indicated by the dotted lines in Figures 1 to 4 and Figure 6. The distribution of the controls is shown in the Appendix A, as well as uncropped full-width images of the membranes (Appendix A). The patients study protocol was approved by the local ethics committees and the donor hearts were approved by the University of Sydney (HREC #7326). All the collections were made with written informed consent obtained from each patient prior to surgery and from the donors’ next of kin.

### 2.2. Whole Protein Isolation

Frozen tissue was dissected on dry ice and homogenized with RIPA buffer (0.05 M C_4_H_11_NO_3_·HCl pH 8.0, 0.15 M NaCl, 1% IGEPAL CA-630, 0.5% sodium deoxycholate, 1% sodium dodecyl sulfate) using a Tissuelyser II (Qiagen). The lysates were centrifuged and the supernatant was collected and passed through an insulin syringe (Becton Dickinson Microlance needle, 25 gauge). Whole protein lysate was mixed with the 4× Laemmli sample buffer (8% sodium dodecyl sulfate, 40% glycerol, 0.04% bromophenol blue, 0.240 M C_4_H_11_NO_3_·HCl pH 6.8, 10 mM β-mercaptoethanol, 0.01 M NaF, 0.001 M Na_3_VO_4_, cOmplete mini protease inhibitor cocktail (Roche)), boiled for 5 min and stored at −80 °C.

### 2.3. Electrophoresis and Western Blots 

Equal amounts of protein (10 µg) were separated on pre-cast SDS-PAGE 4–12% criterion gels (Bio-Rad) and transferred onto nitrocellulose membranes (Bio-Rad). The membranes were blocked in 5% skim milk/1× TBST for 1 h at room temperature and incubated overnight at 4 °C with the following primary antibodies in 3% BSA/1× TBST: mouse anti-acetylated α-tubulin (T7451, Sigma-Aldrich, Saint Louis, MO, USA, mouse anti-α-actinin (A7811, Merck KGaA, Darmstadt, Germany), mouse anti-HSPA1 (ADI-SPA-810, Enzo Life Sciences, Zandhoven, Belgium), mouse anti-HSPA2 (66291-1, Proteintech Group, Rosemont, IL, USA), mouse anti-HSPB1 (ADI-SPA-800, Enzo Life Sciences, Zandhoven, Belgium), rabbit anti-HSPB5 (ADI-SPA-223, Enzo Life Sciences, Zandhoven, Belgium), rabbit anti-HSPB7 (ab150390, Abcam, Cambridge, UK), rabbit anti-HSPD1 (ADI-SPA-805, Enzo Life Sciences, Zandhoven, Belgium), mouse anti-GAPDH (10R-G109a, Fitzgerald Industries International, Acton, MA, USA), mouse anti-LC3B (#2775, Cell Signaling Technology, Danvers, MA, USA), rabbit anti-p62 (#5114, Cell Signaling Technology), mouse anti-α-tubulin (T9026, Sigma-Aldrich), and rabbit anti-ubiquitin (#3933, Cell Signaling Technology). The membranes were incubated for 1 h at room temperature with horseradish peroxidase-conjugated secondary antibody (DakoCytomation, Santa Clara, CA, USA), raised in goat, in 3% BSA/1x TBST. Signals were detected by Amersham Imager 600 (GE Healthcare, Chicago, IL, USA) and quantified by densitometry (ImageQuant TL, GE Healthcare). To correct for loading differences, protein amounts were expressed relative to GAPDH.

### 2.4. Statistical Analyses

Data were analyzed with SPSS Statistics version 22.0 for Windows (IBM Corporation, Armonk, NY, USA) and GraphPad Prism version 7.0 (GraphPad Software Inc., San Diego, CA, USA). Data in figures are presented as means ± standard errors of the mean (SEM) per group and in the text, data are presented as means ± standard deviations (SD). The distribution of each data set was determined by Q-Q plots and then double checked by Kolmogorov-Smirnov and Shapiro Wilk tests after data collection was completed. Normally distributed data were analyzed with unpaired *t*-test (comparing 2 groups) and ordinary one-way ANOVA (comparing > 2 groups) with Tukey’s multiple comparisons post-hoc test (comparing ≤ 3 groups) and Dunnett’s multiple comparisons post-hoc test (comparing > 3 groups). To analyze non-parametric data, the Mann-Whitney U test (comparing 2 groups) and the Kruskal-Wallis test with Dunn’s multiple comparisons post-hoc test (comparing > 2 groups) were used. A two-sided *p* < 0.05 was considered to indicate statistical significance. Multivariate generalized linear model testing (detailed description in the Appendix A) was performed to study the main effects of different mutation groupings, sex, and age at operation on PQC proteins, which were used as dependent variables.

## 3. Results

### 3.1. Changes of Protein Quality Control in HCM_SMP_ and HCM_SMN_

We investigated the expression levels of key PQC proteins to determine if they were altered in patients with HCM at the time of myectomy compared with the levels in LV samples from healthy controls. As the mutant sarcomere proteins may directly affect PQC, we used Western blots to compare myectomy samples from 38 HCM_SMP_ and 14 HCM_SMN_ patients with LV tissue from nine healthy controls that were age- and sex-matched to the HCM samples (Table 1).

#### 3.1.1. HSPs that Stabilize Target Proteins

Dependent on their function, HSPs were classified in two categories: stabilizing HSPs and HSPs with refolding capacity. Stabilizing HSPs (HSPB1 (also known as HSP27), HSPB5 (αB-crystallin), HSPB7 (cardiovascular HSP)) have relatively low molecular weights (15–30 kDa). They stabilize misfolded proteins using an ATP-independent holdase activity, thereby preventing their aggregation (Figure 1A). Compared to controls, the HCM_SMP_ samples expressed significantly higher levels of HSPB1 and HSPB7, while HSPB5 appeared to be unaltered (Figure 1C; Table 2). 

We then quantified the protein levels of these stabilizing HSPs (HSPB1, HSPB5 and HSPB7) in HCM_SMN_ samples. Only HSPB7 was significantly higher in HCM_SMN_ compared to controls (Figure 1C; Table 2). Despite this, there were no significant differences in stabilizing HSPs levels between HCM_SMP_ and HCM_SMN_.

#### 3.1.2. HSPs That Refold Target Proteins

Next we investigated the expression levels of HSPD1 (HSP60), HSPA1 (HSP70 and HSP72) and HSPA2 (HSP70-2) that have an ATP-dependent folding activity. These HSPs have molecular weights >30 kDa and either refold misfolded proteins or assist in their degradation. HSPD1 is mainly located in mitochondria while HSPA1 and HSPA2 are cytosolic, but when they are exposed to heat stress, both translocate to the nucleus [60,61]. Compared to controls, significantly higher levels of HSPD1 and HSPA2 were observed in HCM_SMP_ samples, while HSPA1 levels were not significantly changed. Levels of HSPs with a refolding capacity were not significantly different in HCM_SMN_ compared to controls, although HSPD1 and HSPA2 showed a trend to higher levels. There were no significant differences in HSPs with a refolding capacity between the two HCM groups (Figure 2; Table 2).

#### 3.1.3. Degradation of Proteins

PQC is also involved in the degradation of proteins and therefore levels of ubiquitination, p62 and LC3BII were assessed. The antibody used for ubiquitin recognizes polyubiquitinated proteins without distinguishing between K48-linked polyubiquitin chains (predominantly targeted by proteasomal degradation) and K63-linked polyubiquitin chains (targeted by autophagic degradation) [62]. Levels of ubiquitin and the ubiquitin-binding autophagic adaptor protein p62 were unaltered in both HCM groups compared to controls. Levels of the autophagosome marker LC3BII were significantly higher in HCM_SMP_ compared to controls, while only some samples of the HCM_SMN_ group had higher levels of LC3BII (Figure 3, Table 2).

To ensure that the significant differences detected by ANOVA testing were not related to differences in age and/or sex of the myectomy patients, we performed multivariate testing (Wilks’ Lambda test). This test uses age, sex and the HCM grouping (controls, HCM_SMP_ and HCM_SMN_) as the effects and the HSPs, ubiquitination, p62, and LC3BII as dependent variables. We found an overall significant difference in the levels of key PQC players based on the different mutation groups (Table 3), while sex and age at operation had no significant effect on levels of key PQC players. To determine which of the individual key PQC players differed among groups when taking into account sex and age, we performed tests for between-subjects effects. This test revealed statistically significant differences among groups for HSPB1 and HSPD1 (Table 4). 

Since we found between-subjects main effects, we performed a contrast analysis to determine significant pairwise differences between controls and the two HCM patient groups corrected for age and sex. A simple main effects analysis revealed significantly higher levels of HSPB1, HSPD1 and HSPA2 in HCM_SMP_ compared to controls (Table 5), while a trend to higher levels of HSPD1 was observed in HCM_SMN_. 

Overall, our analyses showed increased levels of several HSPs in HCM_SMP_ and HCM_SMN_ compared to controls, while no significant differences between HCM_SMP_ and HCM_SMN_ were observed.

### 3.2. Changes of Protein Quality Control: HCM Disease Mechanism

All the studied HCM_SMP_ patients were heterozygous for the mutation, implying the potential to produce either or both the mutant and wild-type protein [14]. Either the expression of the mutant protein, which above a threshold (the ‘poison polypeptide’ mechanism) level might become toxic or have an inability to upregulate the haplotype protein levels to meet the needs of the heart (the ‘haploinsufficiency’ mechanism) might underlie HCM [14]. Truncation and missense *MYBPC3* mutations usually involve haploinsufficiency as the disease mechanism since total cMyBP-C protein level is lower and the mutant protein has not been detected in human myectomy samples [13,63]. 

We determined whether key PQC players were differently changed in samples with haploinsufficiency (HI) and poison polypeptide (PP) by comparing 19 *MYBPC3* mutation samples (HCM_HI_) with 19 HCM_SMP_ samples carrying a missense mutation (HCM_PP_). Proteomics analysis using mass spectrometry revealed significantly lower cMyBP-C protein levels in the HCM_HI_ samples compared to the HCM_PP_ patients, consistent with a protein haploinsufficiency mechanism (Figure 4) [64]. No significant differences in PQC components were observed between HCM_HI_ and HCM_PP_ (Figure 5, Table 6).

### 3.3. Microtubule Network

Microtubule-based transport of proteins is required for autophagy and is thus an essential part of PQC. Acetylation of α-tubulin is associated with stable microtubules and is used as a marker of their stability [65,66]. Mouse models of familial HCM showed higher levels of acetylated α-tubulin [67]. Compared to controls, α-tubulin expression levels were significantly higher in both HCM_SMP_ and HCM_SMN_ (Figure 6A,B; Table 2). In addition, acetylation of α-tubulin was significantly higher in both HCM patient groups compared to controls (Figure 6A,B). The increases in the levels of α-tubulin and acetylated α-tubulin were significantly higher in HCM_SMP_ compared to HCM_SMN_.

When comparing mutation types, significantly higher levels of α-tubulin and acetylated α-tubulin were observed in HCM_HI_ compared to HCM_PP_ (Figure 6C, Table 6). The levels of acetylated α-tubulin significantly correlated with α-tubulin (Figure 6D), which is in line with the observation that acetylation of α-tubulin compartmentalized on the stable microtubules. Moreover, a significant inverse linear correlation was present between the cMyBP-C protein level and α-tubulin in HCM with a sarcomere gene mutation (Figure 6E). Overall, these data show that the presence of a sarcomere mutation coincides with a more extensive and stabilized microtubule network in HCM. 

### 3.4. Strong Correlation of HSPs with α-tubulin

Since the PQC is strongly linked to the microtubules network, we performed multivariate testing (Wilks’ Lambda test) using levels of α-tubulin, sex, and age at operation as effects and the HSPs, ubiquitination, p62, and LC3BII as dependent variables. There was an overall statistically significant difference in the levels of key PQC players based on the α-tubulin levels (Table 7). 

To determine which of the key PQC players differed when α-tubulin levels changed, we performed tests for between-subjects effects. This test revealed that α-tubulin had a significant effect on all the measured HSPs, but not on the degradation markers (Table 8). The correlation of HSPs and α-tubulin levels is visualized in Appendix A.

## 4. Discussion

The incomplete penetrance, age-related onset and the large clinical variability in disease severity imply a complex HCM pathophysiology mechanism. Patients with a sarcomere mutation have a particularly higher risk for adverse outcomes. The underlying causes for the difference in disease between HCM_SMP_ and HCM_SMN_ remain unclear. 

Here, we determined the protein levels of key PQC players, including HSPs and degradation markers, and the acetylation of α-tubulin by Western blot analysis in a large set of cardiac samples from HCM_SMP_ and HCM_SMN_ patients to define PQC changes in HCM at the time of myectomy, and identified if the presence of a sarcomere mutation mediates changes in PQC mechanisms and tubulin network. One of the limitations of the current study was that patients had a severe phenotype with LVOT obstruction. Therefore, this group may not be representative for all patients with HCM.

The main findings of our PQC assessment were that: (1) several HSPs and autophagy markers were higher in HCM compared to controls, but there are no major differences between HCM_SMP_ and HCM_SMN_ or mutation type (haploinsufficiency versus poison polypeptide) at the time of myectomy, (2) after correction for sex and age at operation, the major increases in HCM_SMP_ compared to controls were observed in the stabilizing HSPB1 and the refolding HSPD1 and HSPA2, (3) the most significant increase compared to control samples was observed in the levels of α-tubulin and acetylated α-tubulin, with significant differences between the mutation groups: HCM_HI_ > HCM_PP_ > HCM_SMN_, and (4) the levels of α-tubulin significantly correlated with the levels of acetylated α-tubulin and all the HSPs (Figure 7).

Overall, our analyses indicate that carrying a HCM-causing mutation affects PQC and α-tubulin acetylation. We propose that reduced levels of cMyBP-C caused by *MYBPC3* mutations trigger HSPs and α-tubulin acetylation, and may present a pathomechanism in cMyBP-C haloinsufficiency-mediated HCM. The possible impact of the observed PQC changes is discussed below.

### 4.1. Elevated Levels of HSPs in Human HCM

We identified higher levels of HSPB1, HSPD1, and HSPA2 in HCM_SMP_ compared to controls, which could not be explained by differences in sex and age at operation (Table 5). Our findings are in line with studies in human end-stage heart failure studies samples, which showed increases in HSPD1 and HSPB1 compared to controls [68]. 

HSPB1 is implicated in different cardio-protective processes involving interception of misfolded proteins, cytoskeletal organization, anti-apoptotic and anti-oxidant properties [69]. Therefore, its expression may improve the resistance of tissue exposed to stress and injuries [70]. While moderate cardiac expression of HSPB1 protects against doxorubicin-induced cardiac dysfunction through anti-oxidative stress, reports on the effects of HSPB1 overexpression have been inconsistent [71]. Overexpression of HSPB1 conferred protection against myocardial ischemia-reperfusion injury [72], whereas overexpression of HSPB1 caused reductive stress and cardiomyopathy in another study [73]. Further experiments are needed to clarify if the observed increase in HSPB1 is actually beneficial and may be a response to stress exposure.

HSPD1 fulfills multiple tasks in a cell by refolding key proteins after their import into the mitochondria, transporting proteins between the mitochondrial matrix and the cytoplasm of the cell, and binding to cytosolic pro-apoptotic proteins to sequester them [74]. Upregulation of HSPD1 due to mitochondrial impairment is suggested to be an indicator of mitochondrial stress [75]. A total of 15–20% of HSPD1 is located at extra-mitochondrial sites, suggesting important physiological roles at other locations [61]. In human end-stage heart failure, HSPD1 localizes to the plasma membrane, thereby losing anti-apoptotic and protective effects and becoming detrimental to the cell [76]. Impairment of HSPD1-mediated anti-stress response has been reported in dilated cardiomyopathy (DCM) which progressed to end stage heart failure [77]. Here, we identified higher levels of HSPD1, which may indicate mitochondrial stress, as has been reported in HCM [78]. 

The Western blot analysis showed higher levels of HSPA2 in HCM_SMP_ compared to controls. In our mass-spectrometry-based proteomics study, HSPA2 was highly upregulated in cardiac disease (Figure 8A) [64]. HSPA2 has been reported in heart tissue and low expression was identified in fibroblasts [60,79]. Recently, HSPA-family chaperones have been suggested to interact with cMyBP-C to regulate its turnover [80]. Moreover, a significant inverse linear correlation was present between cMyBP-C protein levels and HSPA2 (Figure 8B). Therefore, HSPA2 may be an additional regulator of cMyBP-C. Further experiments are warranted to explain the role of HSPA2 in haploinsufficiency for cMyBP-C.

In contrast, the levels of the HSPB5 and HSPA1 in HCM samples were not different compared to controls after adjusting for sex and age differences. Several studies have shown that HSPB5 and HSPA1 are both increased early in response to acute hypertrophic signaling, but with opposite functions. While HSPB5 is involved in counteracting acute cardiac remodeling by suppressing hypertrophic signaling pathways, HSPA1 supports pro-hypertrophic signaling pathways [81,82]. The increase of HSPA1 was transient and attenuated 14 days after isoproterenol-infusion in a mouse model for cardiac hypertrophy [82]. Consistent with our findings in myectomy samples from patient with advanced disease, HSPA1 levels were not changed in septal myectomy samples of HCM patients and end-stage heart failure patients compared to controls [68,80]. Increased HSPA1 in the heart protects against acute cardiac stress, but not against chronic cardiac stress [83]. 

Based on these studies, we propose that HSPB5 and HSPA1 levels were not changed in the HCM myectomy samples because of persistent chronic hypertrophic stimuli leading to their blunted response. Further research is warranted to investigate if the failure to increase HSPB5 implies a loss of responsiveness at advanced disease stage because HSPB5 seems to play a beneficial role in hypertrophic signaling.

### 4.2. Protein Degradation in HCM

Defects in the turnover of key sarcomere proteins may account for both hypertrophic and functional defects [9]. UPS dysfunction is characterized by the accumulation of ubiquitinated proteins, altered proteasome activity, and changes in expression of UPS proteins or E2 and E3 enzymes [84]. Upon correction for sex and age, no significant changes were observed in the levels of degradation-associated markers in HCM, such as HSPB7, ubiquitin, p62 and LC3BII. Compared to the protein folding-supportive HSPB1 and HSPB5, HSPB7 plays a critical role in preventing protein aggregation [85]. Genetic variants in *HSPB7* have been associated with advanced heart failure and systolic dysfunction [86]. In a zebrafish model, HSPB7 was involved in early post-damage processing of large cytoskeletal proteins and the loss of HSPB7 function was accompanied by an increased damaged protein load, protein aggregation and a risk of cardiomyopathy [87]. Due to the use of an antibody recognizing both K48- and K63-linked polyubiquitinated chains, we do not know if the ratio between K48- and K63-linked polyubiquitinated proteins was altered in HCM. Therefore, we cannot draw any conclusion about the functionality of the proteasome, which would be indicated by an increase in K48-linked polyubiquitinated proteins. LC3BII and p62 are both markers for autophagosomes because in selective autophagy, the cargo adapter protein p62 links polyubiquitinated proteins to the autophagic machinery via LC3 [88]. We did not find a change in autophagosome markers, but this finding does not exclude the prospect that the autophagic flux might be impaired. Autophagy has been implicated in the pathogenesis of a wide range of cardiac pathologies [62,88]. Pre-clinical models of pressure overload-induced cardiac hypertrophy showed that the increase of autophagic flux correlated with the degree of hypertrophy and therapeutic inhibition of autophagy was accompanied by reduced amount of fibrosis [89].

Our analysis of protein degradation markers indicates that there is no enhancement of protein degradation pathways at the time of myectomy. This may indicate that boosting protein degradation may actually represent a treatment option to prevent cardiac dysfunction and remodeling.

### 4.3. Increased Mutation-Dependent Proliferation and Acetylation of α-tubulin

Our protein analyses revealed significantly increased levels of α-tubulin in both HCM groups (Figure 6A), which correlated with an increase in α-tubulin acetylation and the levels of all HSPs. Changes of microtubules distribution occurred at the onset of hypertrophy [90]. In chronically hypertrophied hearts, the increase of microtubules has been shown to be a major cause of contractile defects [91]. Two different hypotheses may explain microtubule densification: (1) microtubules may be involved in myofibrillogenesis and an increase in microtubules would be required when new myofibrils are formed in hypertrophied cardiomyocytes, and (2) microtubules may be intracellular compression-bearing elements needed to cope with increased contractile stress on cardiomyocytes [92,93]. In failing human hearts, the accumulation of microtubules impedes sarcomere motion and contributes to a decreased ventricular compliance [94]. Our results are in line with literature, showing an increase in α-tubulin in the HCM myocardial tissue [95]. The underlying pathomechanism seems to have an effect on extent of increase in α-tubulin and acetylated α-tubulin, since we observed the largest increase in samples with haploinsufficiency for cMyBP-C. 

An intact microtubule-based cytoskeleton contributes to the stabilization of myofibril structure in the cardiomyocytes and HSPs may preserve this relationship [96]. Post-chaperonin tubulin-folding cofactors, such as HSPs and other tubulin-associated proteins, regulate the synthesis, transport, and storage of α- and β-tubulin [97]. For instance, HSPA1 binds to native tubulin dimers and microtubules to assist in proper tubulin folding [98]. In addition, HSPB5 prevents the aggregation of tubulin and the chaperone activity is accompanied by the formation of large complexes between HSPB5 and tubulin [99]. We identified that α-tubulin level significantly correlates with all the investigated HSPs. 

In addition to HSPs, acetylation of α-tubulin was higher in both HCM groups (Figure 6B). Acetylation of α-tubulin directly tunes the compliance and resilience from mechanical breakage to ensure the persistence of long-lived microtubules [100]. Moreover, acetylation of α-tubulin most often compartmentalizes on the stable microtubules and is suggested to increase the assembly of autophagic cargo along microtubules, thereby initiating augmented autophagic degradation in the heart [101,102]. Interestingly, there is a strong linear relation between acetylated and total α-tubulin levels (Figure 6D). This may suggest hyperacetylation of α-tubulin, which is a common response to several cellular stresses. For example, acetyltransferase induction is triggered by the release of mitochondrial reactive oxygen species (ROS) and by AMP kinase [103,104]. Noteworthily, the underlying pathomechanism (HI versus PP) seems to have an effect on the extent of the increase in α-tubulin and the acetylation of α-tubulin, since we observed the largest increase in samples with haploinsufficiency for cMyBP-C. As such, our analyses suggest that proliferation of the microtubular network represents a pathomechanism in HI-mediated HCM. Based on previous studies [100,101,102,103,104], we propose that the increased tubulin network may function to maintain cellular stability and enhance autophagic flux of damaged proteins, while it may impair contractile function [95,105]. Future studies are warranted to establish this link (cause-consequence) between reduced cMyBP-C protein levels, increased α-tubulin acetylation and cardiomyocyte contractile function and hypertrophy that may underlie initiation and progression of cardiac disease in *MYBPC3* mutation carriers.

### 4.4. Inter-Individual Differences

We observed large inter-individual differences in various key PQC players and tubulin networks, especially in the presence of a sarcomere mutation. This variability may be explained by the large variability among HCM-causing mutations, which may all have a distinct effect on cellular PQC. Most of the mutation sites of proteins affect stability and aggregation but rarely its function [106]. This concept is supported by the fact that HCM_HI_ mutations cause a rather heterogeneous cMyBP-C haploinsufficiency, with some samples having relatively low cMyBP-C, and others having still relatively high protein level. The loss of cMyBP-C levels correlates very well with α-tubulin, thereby supporting the diverse response to a change in sarcomere protein composition in the case of HCM_HI_. The microtubule response correlates significantly with the expression level of cMyBP-C, and may represent a direct compensation for the loss of cMyBP-C as cMyBP-C is known to have a central regulatory role in myofibril development and stability [107]. In line with this, reduced myofibril density was detected in septal myectomy samples from HCM_SMP_ patients compared to HCM_SMN_ patients [18]. While the excessive microtubular response to cMyBP-C haploinsufficiency may be an attempt to maintain myofibril stability, it may aggravate cardiac dysfunction by altering cytoskeletal mechanical properties [95,105]. 

## 5. Conclusions

In summary, our data show that the hypertrophied muscle of HCM patients is characterized by an increased and stabilized tubular network which may be an adaptive response to cardiomyocyte growth. While this compensatory response may aid to stabilize cardiomyocyte architecture, this structural adaption may contribute to a reduced function of the HCM myocardium.

## Figures and Tables

**Figure 1 cells-08-00741-f001:**
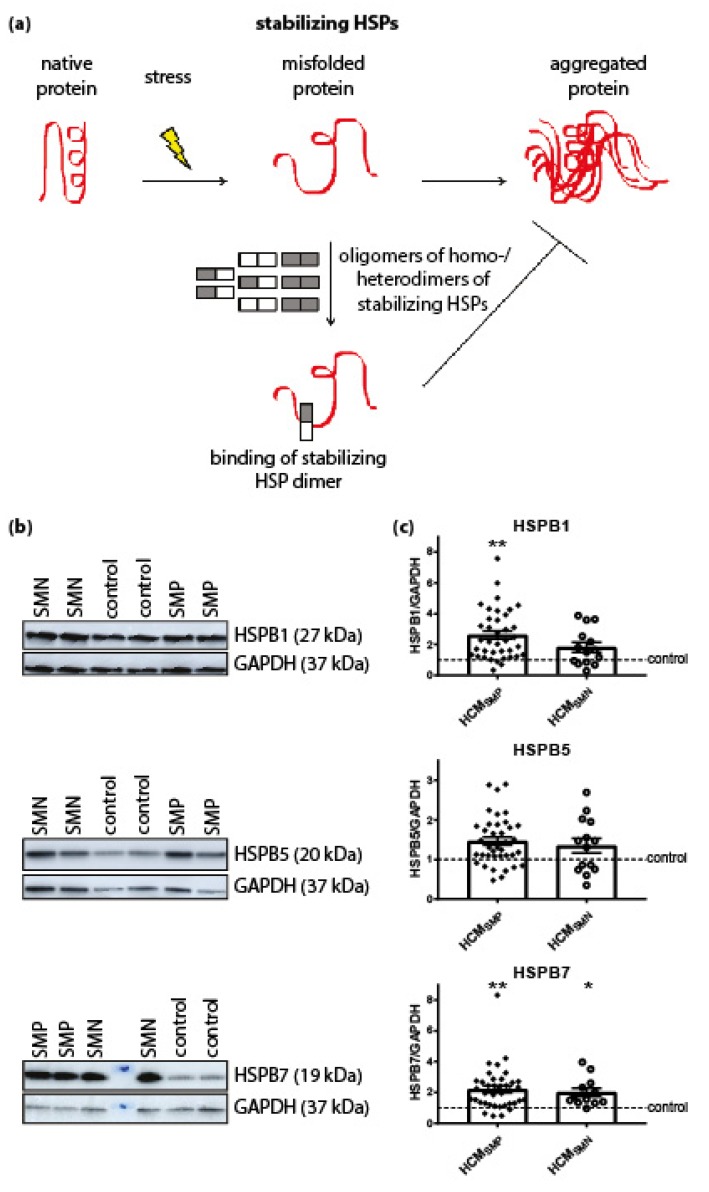
Stabilizing HSPs. (**a**) Under stress conditions when native proteins are destabilized and begin to unfold, stabilizing HSPs dissociate from homo- and/or heterogeneous oligomeric complexes into dimers to bind these partially misfolded proteins. Thereby, stabilizing HSPs prevent the aggregation of misfolded proteins. Figure reproduced and adapted with permission from Dorsch, L.M. et al., Pflügers Archiv–European Journal of Physiology; published by Springer Berlin Heidelberg, 2018. (**b**) Representative blot images for HSPB1, HSPB5, and HSPB7 expression. (**c**) Higher levels of HSPB1 and HSPB7 in HCM_SMP_ (*n* = 38) compared to controls. Higher HSPB7 levels in HCM_SMN_ (*n* = 12) compared to controls (*n* = 9; average is shown as dotted line). HSPB1 and HSPB5 protein levels were not different in HCM_SMP_ (*n* = 14) compared to controls. There were no significant differences between HCM_SMP_ and HCM_SMN_. Each dot in the scatter plots represents an individual sample. * *p* < 0.05 and ** *p* < 0.01 versus controls.

**Figure 2 cells-08-00741-f002:**
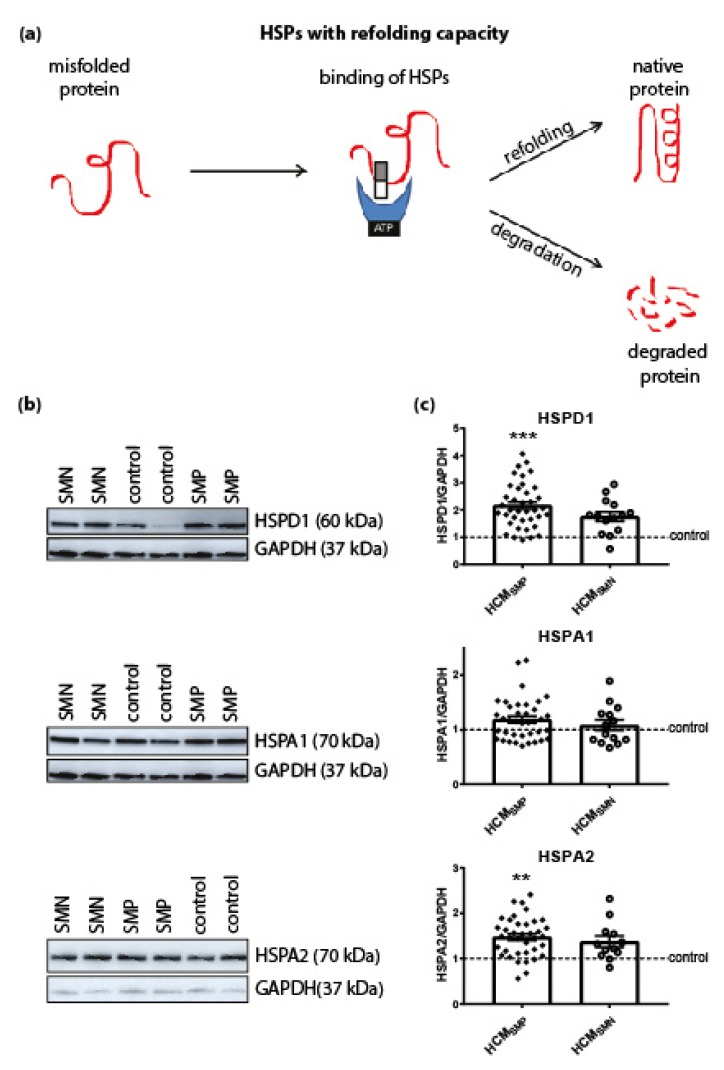
HSPs with a refolding capacity. (**a**) HSPs with ATPase activity and stabilizing HSPs bind to the misfolded protein to stabilize and further process it. HSPs with a refolding capacity either refold the misfolded protein to its native structure or, if refolding is impossible, the HSPs assist in the degradation pathways to degrade the misfolded protein. Figure reproduced and adapted with permission from Dorsch, L.M. et al., Pflügers Archiv—European Journal of Physiology, published by Springer Berlin Heidelberg, 2018. (**b**) Representative blot images for HSPD1, HSPA1, and HSPA2 expression. (**c**) Higher levels of HSPD1 and HSPA2 in HCM_SMP_ (*n* = 38) compared to controls. HSPs with refolding capacity were unaltered in HCM_SMN_ (HSPD1, HSPA1: *n* = 14; HSPA2: *n* = 12) compared to controls. There were no significant differences between HCM_SMP_ and HCM_SMN_. Each dot in the scatter plots represents an individual sample. ** *p* < 0.01 and *** *p* < 0.001 versus controls.

**Figure 3 cells-08-00741-f003:**
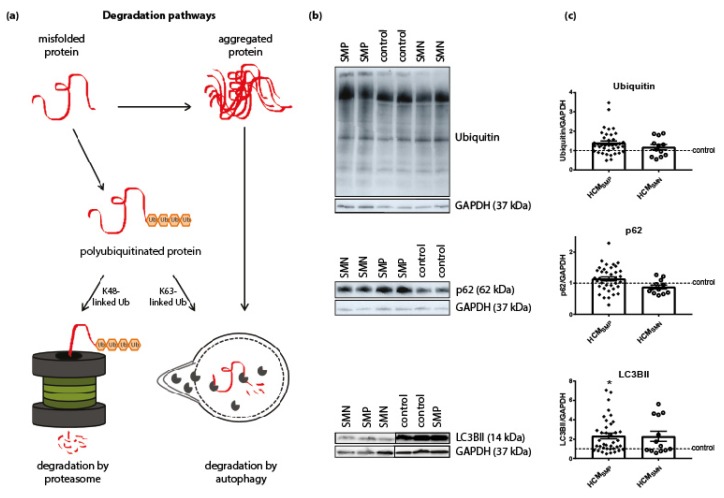
Degradation pathways. (**a**) Misfolded proteins with polyubiquitin chains linked via lysine 48 (K48) are mainly degraded by the proteasome. Misfolded proteins carrying K63-linked polyubiquitin chains and aggregated proteins enter the autophagic pathway. Figure reproduced and adapted with permission from Dorsch, L.M. et al., Pflügers Archiv—European Journal of Physiology; published by Springer Berlin Heidelberg, 2018. (**b**) Representative blot images for ubiquitin, p62, and LC3BII expression. (**c**) Protein levels of ubiquitin and p62 did not differ between HCM and controls samples. Higher levels of LC3BII in HCM_SMP_ (*n* = 38) compared to controls. There were no significant differences between HCM_SMP_ and HCM_SMN_ (ubiquitin, p62: *n* = 12; LC3BII: *n* = 13). Each dot in the scatter plots represents an individual sample. * *p* < 0.05 versus controls.

**Figure 4 cells-08-00741-f004:**
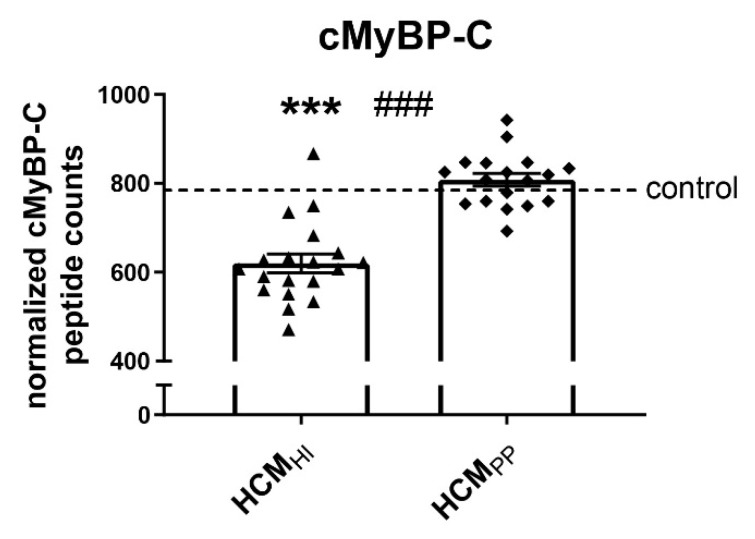
cMyBP-C haploinsufficiency. Mass spectrometry-based proteomics analysis revealed lower levels of cMyBP-C in HCM_HI_ (*n* = 19) compared to controls (*n* = 5) and HCM_PP_ (*n* = 18). Each dot on the scatter plot represents an individual sample. *** *p* < 0.001 versus controls and ^###^
*p* < 0.001 versus HCM_PP_.

**Figure 5 cells-08-00741-f005:**
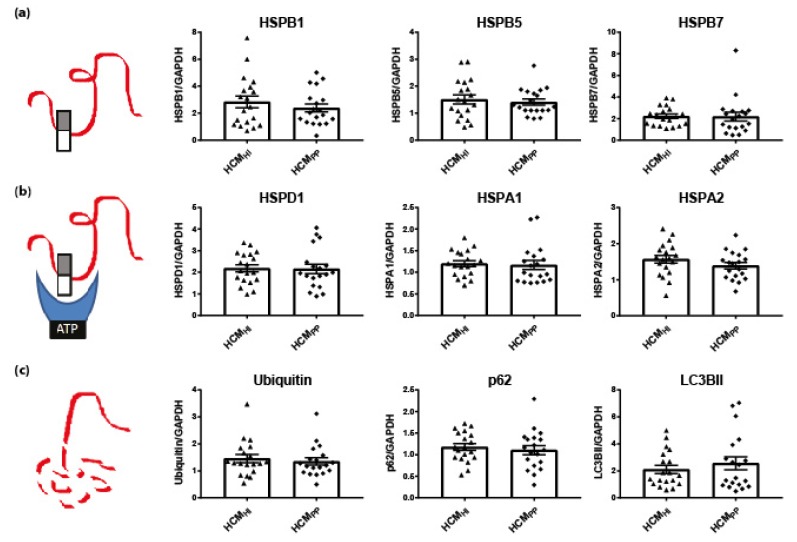
Pathomechanism of HCM and key PQC players. (**a**) Stabilizing HSPs (HSPB1, HSPB5, HSPB7), (**b**) HSPs with refolding capacity (HSPD1, HSPA1 HSPA2) and (**c**) degradation markers (ubiquitin, p62, LC3BII) were not significantly different between HCM_HI_ (*n* = 19) and HCM_PP_ (*n* = 19). Each dot in the scatter plots represents an individual sample. Figure reproduced and adapted with permission from Dorsch, L.M. et al., Pflügers Archiv–European Journal of Physiology; published by Springer Berlin Heidelberg, 2018.

**Figure 6 cells-08-00741-f006:**
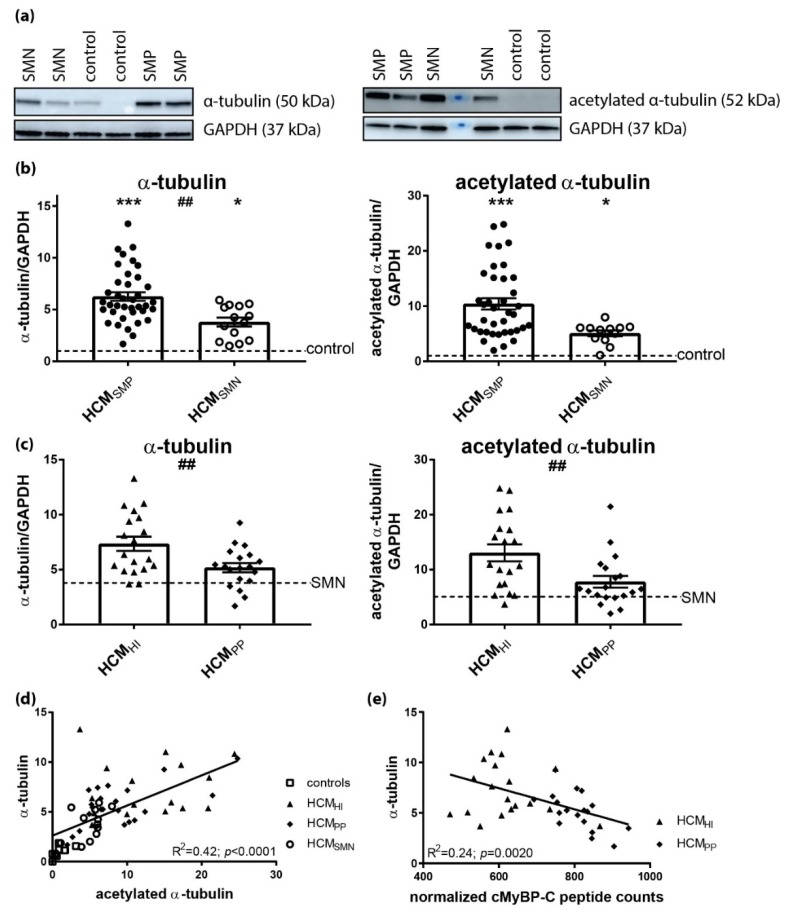
Pathomechanism of HCM and acetylation of α-tubulin. (**a**) Representative blot images for α-tubulin and acetylated α-tubulin. (**b**) Higher levels of α-tubulin and acetylated α-tubulin in HCM_SMP_ (*n* = 38) and HCM_SMN_ (α-tubulin: *n* = 14; acetylated α-tubulin: *n* = 12) compared to controls. Significant difference of α-tubulin levels between HCM_SMP_ and HCM_SMN_ (##p < 0.01). (**c**) Higher levels of α-tubulin and acetylated α-tubulin in HCM_HI_ (*n* = 19) than HCM_PP_ (*n* = 19). (**d**) The levels of α-tubulin correlated well with acetylated α-tubulin (**e**) Significant inverse linear correlation of α-tubulin with cMyBP-C. Controls = open squares, HCM_SMP_ = filled circles, HCM_SMN_ = open circles, HCM_HI_ = filled triangles, HCM_PP_ = filled rhomboids. Each dot in the scatter plots and the correlation analyses represents an individual sample. * *p* < 0.05 and *** *p* < 0.001 versus controls and ^##^
*p* < 0.01 and ^###^
*p* < 0.001 versus HCM_PP_.

**Figure 7 cells-08-00741-f007:**
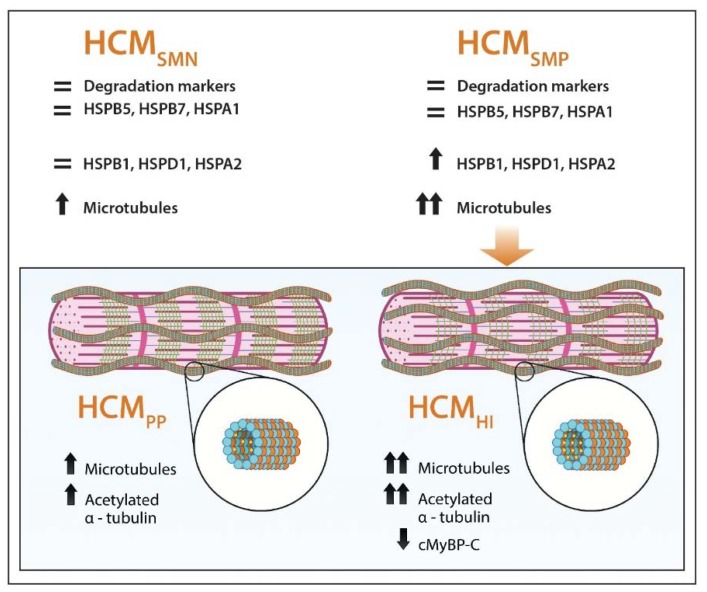
Schematic representation of PQC and microtubular system assessment in HCM. Our analyses show that HSPB1, HSPD1 HSPA2, α-tubulin and acetylated α-tubulin were increased in HCM, especially when a sarcomere mutation is present, while HSPB5, HSPB7, HSPA1 and degradation markers are not changed in HCM compared to controls. Levels of α-tubulin (orange balls) and acetylated α-tubulin (small yellow balls attached to α-tubulin in the lumen of polymerized microtubules (blue and orange balls)) were more increased when haploinsufficiency (HI), characterized by lower levels of cardiac myosin-binding protein-C (green lines), was the underlying pathomechanism instead of poison polypeptides (PP). =: similar to controls; ↑: significantly increased compared to controls; ↑↑: highly significantly increased compared to controls; ↓: significantly decreased compared to controls.

**Figure 8 cells-08-00741-f008:**
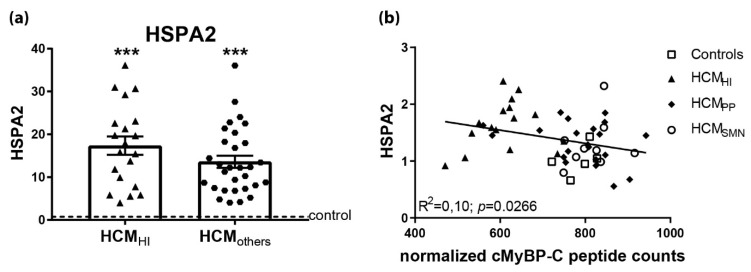
Upregulated HSPA2 in HCM in proteomics screen. (**a**) In a mass-spectrometry based proteomics study, HSPA2 levels were higher in HCM_HI_ (*n* = 20) and HCM_others_ (*n* = 30) samples, including HCM_PP_ and HCM_SMN_, compared to controls (*n* = 8). (**b**) Significant inverse correlation of HSPA2 levels, determined by Western blot analysis with cMyBP-C. Controls = open squares, HCM_HI_ = filled triangles, HCM_PP_ = filled rhomboids, HCM_SMN_ = open circles, HCM_others_ = hexagons. Each dot in the scatter plots and correlation analysis represents an individual sample. *** *p* < 0.001 versus controls.

**Table 1 cells-08-00741-t001:** Characteristics of controls and manifest HCM patients.

Number	Code	Reported Pathomechanism	Affected Gene	Gene/Protein Variant	Type of Mutation	Sex (F/M)	Age	ST (mm)	LVOT (mmHg)
1	HCM 42 [17,18,19,20]	HI	MYBPC3	c.2373dupG	truncation	M	32	23	64
2	HCM 43 [18,19,20,21,22]	HI	MYBPC3	c.2373dupG	truncation	M	60	23	77
3	HCM 103 [20]	HI	MYBPC3	c.2373dupG	truncation	M	26	20	13
4	HCM 104 [20]	HI	MYBPC3	c.2373dupG	truncation	M	33	24	31
5	HCM 120 [20]	HI	MYBPC3	c.2373dupG	truncation	M	27	24	61
6	HCM 123 [20]	HI	MYBPC3	c.2373dupG	truncation	F	59	12	64
7	HCM 169 [20]	HI	MYBPC3	c.2373dupG	truncation	M	52	-	-
8	HCM 34 [18,20,23]	HI	MYBPC3	c.1790G > A; p.Arg597Gln	missense	F	47	20	38
9	HCM 36 [17,18,20,21,22,23,24,25]	HI	MYBPC3	c.927-2A > G	truncation	M	22	30	71
10	HCM 47 [19,20,23,25]	HI	MYBPC3	c.3407_3409delACT	truncation	M	55	25	96
11	HCM 52 [18,19,20]	HI	MYBPC3	c.2827C > T	truncation	F	24	24	34
12	HCM 62 [17,18,20,23]	HI	MYBPC3	c.772G > A; p.Glu258Lys	missense	M	35	27	3
13	HCM 63 [20,23]	HI	MYBPC3	c.2827C > T	truncation	M	33	21	25
14	HCM 71 [20]	HI	MYBPC3	c.2827C > T	truncation	M	49	16	9
15	HCM 82 [20]	HI	MYBPC3	c.2783C > T; p.Ser928Leu	missense	M	71	20	64
16	HCM 113 [17,20]	HI	MYBPC3	c.2827C > T	truncation	M	21	45	27
17	HCM 116 [20]	HI	MYBPC3	c.2827C > T	truncation	F	53	-	-
18	HCM 124 [20]	HI	MYBPC3	c.2827C > T	truncation	M	53	21	41
19	HCM 133 [20]	HI	MYBPC3	c.442G > A	truncation	M	58	-	-
1	HCM 27 [18,19,20,25]	PP	MYH7	c.4130C > T; p.Thr1377Met	missense	F	58	20	100
2	HCM 42B [18,19,20,23,25]	PP	MYH7	c.1816G > A; p.Val606Met	missense	F	46	20	77
3	HCM 80 [20]	PP	MYH7	c.1291G > C and c.4327G > A; p.Val431Leu and p.Asp1443Asn	missense	M	34	17	85
4	HCM 92 [20]	PP	MYH7	c.2685A > C; p.Glen895His	missense	M	66	-	-
5	HCM 106 [20]	PP	MYH7	c.2783A > T; p.Asp928Val	missense	M	35	16	16
6	HCM 114 [20]	PP	MYH7	c.976G > C; p.Ala326Pro	missense	M	69	19	71
7	HCM 119 [20]	PP	MYH7	c.3367G > A; p.Glu1123Gln	missense	M	41	20	81
8	HCM 130 [20]	PP	MYH7	c.976G > C; p.Ala326Pro	missense	M	72	18	27
9	HCM 131	PP	MYH7	c.1987C > T; p.Arg663Cys	missense	F	52	21	41
10	HCM 166 [20]	PP	MYH7	c.2080C > T; p.Arg694Cys	missense	F	66	-	-
11	HCM 55 [18,23,25]	PP	TNNI3	c.433C > T; p.Arg145Trp	missense	M	46	23	100
12	HCM 59 [18,25]	PP	TNNI3	c.433C > T; p.Arg145Trp	missense	M	66	16	67
13	HCM 163	PP	TNNI3	c.433C > T; p.Arg145Trp	missense	F	64	-	-
14	HCM 170	PP	TNNI3	c.433C > T; p.Arg145Trp	missense	F	69	-	-
15	HCM 132	PP	TNNT2	c. 814C > T; p.Arg272Cys	missense	F	15	-	-
16	HCM 173	PP	TNNT2	c.832C > T; p.Arg278Cys	missense	M	58	-	-
17	HCM 175	PP	TNNT2	c.832C > T; p.Arg278Cys	missense	M	61	-	-
18	HCM 135	PP	MYL2	c.64G > A; p.Glu22Lys	missense	M	68	-	-
19	HCM 88	PP	MYL2	c.401A > C; p.Glu134Ala	missense	F	57	-	-
1	HCM 30 [18,19,26]	SMN			-	F	72	24	88
2	HCM 40	SMN			-	M	22	15	100
3	HCM 90	SMN			-	M	53	17	61
4	HCM 96	SMN			-	M	74	17	81
5	HCM 100	SMN			-	F	41	-	81
6	HCM 105	SMN			-	M	65	16	49
7	HCM 109	SMN			-	F	71	20	130
8	HCM 117 [17]	SMN			-	F	50	16	159
9	HCM 125 [17]	SMN			-	M	66	18	132
10	HCM 126	SMN			-	F	64	15	104
11	HCM 156	SMN			-	F	54	-	-
12	HCM 168	SMN			-	M	46	-	-
13	HCM 172	SMN			-	M	57	-	-
14	HCM 174	SMN			-	M	66	-	-
1	3145 [27,28,29,30,31,32,33,34,35,36,37,38,39]	Ctrl			-	M	39	-	-
2	4021 [33,34,35,36,37,38,39,40,41,42,43,44,45]	Ctrl			-	F	53	-	-
3	4049 [33,34,35,36,37,38,39,40,41,42,46,47,48,49,50,51]	Ctrl			-	M	65	-	-
4	5126 [46,47,48,49,52]	Ctrl			-	F	55	-	-
5	6008 [42,53,54,55]	Ctrl			-	M	40	-	-
6	7012 [27,28,29,30,31,32,56,57]	Ctrl			-	M	19	-	-
7	7040 [33,34,35,36,55,58]	Ctrl			-	M	37	-	-
8	7054 [55,59]	Ctrl			-	M	33	-	-
9	8004 [55]	Ctrl			-	F	50	-	-

HI: haploinsufficiency; PP: poison polypeptide; SMN: sarcomere mutation-negative; Ctrl: healthy donor hearts; F: female; M: male; Age at time of surgery; ST: septal thickness; LVOT: left outflow tract pressure gradient at rest. Patients without significant obstruction (LVOT gradient < 20 mmHg), but with severe heart failure symptoms, received an extended myectomy in order to increase LV filling.

**Table 2 cells-08-00741-t002:** Overview of results on statistical testing for differences between controls, HCM_SMP_ and HCM_SMN_.

			Adjusted *p* Value after Post-Hoc Testing		
Proteins	Statistical Test	*p* Value	Controls vs. HCM_SMP_	Controls vs. HCM_SMN_	HCM_SMP_ vs. HCM_SMN_
Stabilizing HSPs					
HSPB1	Kruskal-Wallis	0.0019	0.0018	0.2113	0.3302
HSPB5	Ordinary one-way ANOVA	0.1297	0.1090	0.3807	0.8174
HSPB7	Kruskal-Wallis	0.0060	0.0048	0.0341	>0.9999
Refolding HSPs					
HSPD1	Ordinary one-way ANOVA	0.0004	0.0003	0.0530	0.2123
HSPA1	Kruskal-Wallis	0.3636	0.6382	>0.9999	>0.9999
HSPA2	Ordinary one-way ANOVA	0.0101	0.0071	0.1008	0.7334
Protein Degradation					
Ubiquitin	Kruskal-Wallis	0.2159	0.3094	>0.9999	0.9890
p62	Ordinary one-way ANOVA	0.1087	0.5822	0.7531	0.1046
LC3BII	Kruskal-Wallis	0.0451	0.0384	0.2455	>0.9999
Tubulin network					
α-tubulin	Ordinary one-way ANOVA	<0.0001	<0.0001	0.0119	0.0019
Acetylated α-tubulin	Kruskal-Wallis	<0.0001	<0.0001	0.0451	0.0516

*p* < 0.05 is considered to be significant.

**Table 3 cells-08-00741-t003:** Multivariate Tests ^a^: HCM grouping (controls, HCM_SMP_, and HCM_SMN_), sex and age at operation.

Effect	Value	F	Hypothesis df	Error df	Significance	Partial eta Squared
Intercept	0.260	14.561 ^b^	9.000	46.000	<0.001	0.740
HCM grouping	0.479	2.278 ^b^	18.000	92.000	0.006	0.308
Sex	0.754	1.664 ^b^	9.000	46.000	0.126	0.246
Age at operation	0.717	2.016 ^b^	9.000	46.000	0.059	0.283

^a^ Design: Intercept and HCM grouping and sex and age at operation; ^b^ exact statistic; F: F statistic for the given effect and test statistic; Hypothesis df: Number of degrees of freedom in the model; Error df: Number of degrees of freedom associated with the model errors; Partial eta squared: estimate of effect size; *p* < 0.05 is considered to be significant.

**Table 4 cells-08-00741-t004:** Tests for between-subjects: HCM grouping (controls, HCM_SMP_, HCM_SMN_) as an effect and correction for differences in sex and age at operation.

	Type III Sum of Squares	df	Mean Square	F	Significance	Partial eta Squared	R squared
Stabilizing HSPs							
HSPB1	23.286	2	11.643	6.0313	0.004	0.182	0.261
HSPB5	1.777	2	0.889	2.495	0.092	0.085	0.088
HSPB7	9.725	2	4.863	3.319	0.044	0.109	0.162
Refolding HSPs							
HSPD1	11.104	2	5.552	10.543	<0.001	0.281	0.323
HSPA1	0.258	2	0.129	0.981	0.381	0.035	0.055
HSPA2	1.724	2	0.862	5.114	0.009	0.159	0.171
Protein degradation							
Ubiquitin	1.433	2	0.716	2.101	0.132	0.072	0.097
p62	0.733	2	0.367	2.498	0.092	0.085	0.122
LC3BII	12.839	2	6.420	2.276	0.112	0.078	0.111

df: Number of degrees of freedom in the model; F: F statistic for the given effect and test statistic; Partial eta squared: estimate of effect size; R squared: proportion of the variance in the dependent variable that is predictable from the independent variable; Bonferroni correction: statistical significance at *p* < 0.0056 (α/9 = 0.05/9 = 0.0056).

**Table 5 cells-08-00741-t005:** Simple Contrast Results (K Matrix) for HCM grouping (controls, HCM_SMP_, HCM_SMN_) as an effect and correction for differences in age at operation and sex.

			Dependent Variable								
			Stabilizing HSPs			Refolding HSPs			Protein degradation		
			HSPB1	HSPB5	HSPB7	HSPD1	HSPA1	HSPA2	Ubiquitin	p62	LC3BII
**HCM_SMP_ vs. controls**	Contrast Estimate		1.804	0.447	1.164	1.242	0.177	0.491	0.441	0.128	1.319
	Standard error		0.520	0.223	0.453	0.271	0.136	0.154	0.218	0.143	0.628
	Significance		0.001	0.050	0.013	<0.001	0.198	0.002	0.049	0.375	0.040
	95% Confidence Interval for	Lower Bound	0.761	−0.00002	0.257	0.698	−0.095	0.183	0.003	−0.159	0.060
		Upper Bound	2.848	0.895	2.072	1.786	0.449	0.799	0.878	0.415	2.578
**HCM_SMN_ vs. controls**	Contrast Estimate		1.522	0.178	0.929	0.961	0.086	0.452	0.301	−0.160	1.335
	Standard error		0.650	0.279	0.565	0.339	0.169	0.192	0.273	0.179	0.784
	Significance		0.023	0.524	0.106	0.006	0.613	0.031	0.274	0.374	0.094
	95% Confidence Interval for	Lower Bound	0.220	−0.380	−0.204	0.282	−0.253	0.041	−0.245	−0.519	−0.236
		Upper Bound	2.825	0.737	2.062	1.641	0.426	0.810	0.848	0.198	2.907

Bonferroni correction: statistical significance at *p* < 0.0028 (α/18 = 0.05/18 = 0.0028).

**Table 6 cells-08-00741-t006:** Overview of results on statistical testing for differences between HCM_HI_ and HCM_PP_.

Proteins	Statistical Test	*p* Value
Stabilizing HSPs		
HSPB1	Unpaired *t*-test	0.4040
HSPB5	Unpaired *t*-test	0.6236
HSPB7	Mann-Whitney U test	0.4181
Refolding HSPs		
HSPD1	Unpaired *t*-test	0.1712
HSPA1	Unpaired *t*-test	0.8044
HSPA2	Unpaired *t*-test	0.2096
Protein degradation		
Ubiquitin	Unpaired *t*-test	0.6099
p62	Unpaired *t*-test	0.5868
LC3BII	Unpaired *t*-test	0.4297
Tubulin network		
α-tubulin	Unpaired *t*-test	0.0074
Acetylated α-tubulin	Unpaired *t*-test	0.0079
Acetylated α-tubulin/α-tubulin	Mann-Whitney U test	0.1629

*p* < 0.05 is considered to be significant.

**Table 7 cells-08-00741-t007:** Multivariate Tests ^a^: α-tubulin (all samples), sex, and age at operation.

Effect	Value	F	Hypothesis df	Error df	Significance	Partial eta Squared
Intercept	0.389	8.186 ^b^	9.000	47.000	<0.001	0.611
α-tubulin	0.427	7.009 ^b^	9.000	47.000	<0.001	0.573
Sex	0.774	1.523 ^b^	9.000	47.000	0.168	0.226
Age at operation	0.791	1.384 ^b^	9.000	47.000	0.223	0.209

^a^ Design: Intercept and α-tubulin and sex and age at operation; ^b^ exact statistic; F: F statistic for the given effect and test statistic; Hypothesis df: Number of degrees of freedom in the model; Error df: Number of degrees of freedom associated with the model errors; Partial eta squared: estimate of effect size; *p* < 0.05 is considered to be significant.

**Table 8 cells-08-00741-t008:** Tests for between-subjects: α-tubulin as an effect and correction for differences in sex and age at operation.

	Type III Sum of Squares	df	Mean Square	F	Significance	Partial eta Squared	R Squared
Stabilizing HSPs							
HSPB1	38.460	1	38.460	23.663	>0.001	0.301	0.368
HSPB5	3.630	1	3.630	11.485	0.001	0.173	0.176
HSPB7	14.246	1	14.246	10.504	0.002	0.160	0.209
Refolding HSPs							
HSPD1	10.273	1	10.273	19.305	>0.001	0.260	0.303
HSPA1	1.370	1	1.370	12.580	0.001	0.186	0.203
HSPA2	1.723	1	1.723	10.410	0.002	0.159	0.171
Protein degradation							
Ubiquitin	0.419	1	0.419	1.187	0.281	0.021	0.047
p62	0.418	1	0.418	2.792	0.100	0.048	0.087
LC3BII	3.849	1	3.849	1.312	0.257	0.023	0.058

df: Number of degrees of freedom in the model; F: F statistic for the given effect and test statistic; Partial eta squared: estimate of effect size; R squared: proportion of the variance in the dependent variable that is predictable from the independent variable; Bonferroni correction: statistical significance at *p* < 0.0056 (α/9 = 0.05/9 = 0.0056).

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
