# Peer review of "Protein Quality Control Activation and Microtubule Remodeling in Hypertrophic Cardiomyopathy"

_cells, 2019, doi:10.3390/cells8070741_

Round 1
Reviewer 1 Report
The manuscript by Drs. Dorsch, van der Velden et al. comprehensively describes the alterations of protein quality control (PCQ) in myectomy samples from patients with hypertrophic cardiomyopathy. The authors successfully identify a link between the presence or absence of specific disease-causing mutations and specific alterations of the different key players of cardiomyocyte proteostasis. In particular, the authors found that stabilizing and refolding heat-shock protein expression is increased in HCM patient samples with sarcomeric mutations, and there is an increase of autophagosome markers, suggesting increase autophagic activity. Sarcomere mutation-positive patient samples were divided based on the site of the causing mutation and the related main patho-mechanism, that is “haploinsufficiency” for truncating or non-sense mutations of MyBPC3 (where a lower level of MyBPC3 is demonstrated) or “poison peptide” for missense mutations of MYH7 or thin-filament protein genes. Interestingly, an increased level of alpha-tubulin and acetylated alpha-tubulin was observed only in patients with MyBPC3 mutations and haploinsufficiency and not in patients with other mutations leading to a “poison-peptide” mechanism of disease. This is highly relevant for the understanding of the pathophysiology of HCM, as microtubule proliferation might be a compensatory mechanism to assure sarcomeric stability when MyBPC is lacking due to haploinsufficiency, with possible consequences on myocardial mechanical function.
The manuscript is very well written, methods and results are clearly presented and discussed. Statistics is sound and took into consideration al possible confounders and biases thanks to a well-performed multivariate analysis. The figures are very clear, and the cartoons that are inserted into each figure are a nice addition that will help the readers understand the role of the proteins investigated and the possible implications related to changes of their levels within the cell.
There are no major issues I have identified in this very good manuscript. However, I have a number of minor observations and suggestions.
1)Table 1.
-Despite the title suggests so, clinical/demographic data from control subjects (donors) are absent in the table. They should be added to the table.
-Four different subgroups of sarcomere-positive HCM patients are clearly separated in the table (MyBPC3 with the Dutch founder-effect mutation, other MyBPC3 mutants, patients with MYH7 mutations and patients with thin-filament mutations); however, in the paper no studies are performed where the four different types of mutations are taken into considerations. Why is that?
-Is LVOT gradient measured during the preoperative echo? If so, why were patients with low gradients (<20mmHg) operated?
2)It is unclear whether the authors tested whether the changes in the levels of stabilizing and refolding HSPs identified in sarcomere-positive patients are different when SMP patients are divided int the two subgroups used for tubulin data (that is, HI vs. PP samples). As tubulin levels are associated with HSP levels, I would expect to observe higher HSPs in haploinsufficiency patients. Is that so?
3)significant changes in HSP levels and autophagy proteins are observed between CTR and sarcomere-positive samples, but are not observed between CTR and sarcomere-negative HCM samples(Table 2). So it appears that these changes are absent in sarcomere-negative samples. However, when SMP are compared with SMN HCM samples, no statistically-significant differences are identified. Why does that happen? Is that because the number of SMN samples is too low?
4)proliferation of microtubules is an intriguing novel patho-mechanism involved in haploinsufficency-mediated HCM, for the possible mechanical consequences. I believe the authors should expand this concept more in the discussion, as it is one of the main results of this paper.
Reviewer 2 Report
The authors have studied a large number of HCM tissue samples and compared them with door heart samples to determine the relationship of the disease to protein quality control systems and microtubules. The results are quite clear, but some details need attention.
Samples, Table 1 additional data needed:
1 for missense mutations give the amino acid change
2 The donor samples come from the Sydney Heart Bank: their code numbers should be included in the table so that correlation can be made by other users of these samples.
3 If any of the myectomies have been used in other published studies, the references should be given.
Statistics. The statistical routines need to be explained and justified in much greater detail (in a supplement if necessary).
1 For instance, in table 2, there seems little justification for using two different statistical tests: if this is based on the observed distribution of data this is obviously incorrect (i.e. one dataset may appear normal but not actually be so if a larger (or a different) dataset was used
2 Need a much fuller explanation of the Wilks' Lambda test and justification for its use. It is very hard to see what is happening here. It would be a great help if some direct plots were shown similar to 6d, e relating measured parameters to age, sex and mutation type.
Define the y axis in figure 4 and the x axis in figure 6e
Figure 7. This figure should be eliminated as uninformative and somewhat misleading. You cannot show three measurement points as a continuous line and the 'microtubule' diagrams are superfluous.
The authors state in the abstract and in the discussion that
Increased tubulin network may function to maintain cellular stability, but it will impair contractile function.
Reference 47 is given, however this paper does not really justify the statement since it does not distinguish between cause and effect for microtubules and contractile function in heart failure. The authors should produce the evidence for this statement or remove it.
Reviewer 3 Report
This study by Dorsch et al. explored the role of protein quality control (PQC) system of cardiomyocytes after Septal Myectomy in Hypertrophic cardiomyopathy (HCM) patients. This is very important study and we need more study like to address major cardiomyopathy-based mortality. In general most of the experiments are well designed however, a general assumption was made that all the sarcomere mutations are limited to PQC. Indeed, patients with a sarcomere mutation have a higher risk for adverse outcomes due to multiple dysregulation but not only limited to PQC. I have following concerns with this manuscript and should be answered in the revised manuscript. Addressing these questions will be very important to address this major disease.
1. HCM linked with mutations of sarcomere proteins are not only limited to defective PQC. In fact, there are multiple factors including calcium handing, ATPase activity, myofibrillar disorganization----etc. However, none of these possibilities were considered/discussed. It is hard to believe all the HCM based diseases are only based upon defective PQC.
2. I am failed to understand how these authors jumped in to microtubules expression without looking at the organization of main sarcomere proteins including myosin, actin, actinin, desmin.
3. As authors also indicated in their hypotheses for microtubule densification; “Microtubules may be involved in myofibrillogenesis and an increase in microtubules would be required when new myofibrils are formed in hypertrophied cardiomyocytes”. However, authors are not looking formation or organization. They should use some cytological/ultrastrastructural approach to address this question.
4. In general, very much entire manuscript is focused on one technique (i.e. immunoblotting) to demonstrate up-or down-regulation of several proteins. This alterations in not enough to address the important questions like protein aggregation or degradation. However, not of these possibilities were considered. Therefore, using some imaging techniques will be certainly helpful.
5. The authors also showed that both autophagy and USP system were altered in with HCM linked mutations. However, their interrelationship is poorly addressed.
6. It is not clear if author run western blots of individuals sample or they used pooled sample?
7. Since, entire manuscript is based upon immunoblotting; the authors should also should uncut and unprocessed images of their western blots in the supplementary section.
8. Entire conclusion was made on one hyothesis (i.e. HCM linked with mutations of sarcomere proteins are and has defective PQC). However, there are other factors also involved with HCM, which were not considered.
9. Most of the conclusion were made based upon up or down regulation of the proteins using immunoblotting, without looking at protein aggregation/degradation.
Round 2
Reviewer 3 Report
My questions/concerns I have raised in previous version has been addressed sufficiently. I do not have any additional questions and this is suitable for publication in Cells. All the best.